POINT OF VIEW

# Tuberculosis innovations mean little if they cannot save lives

**Abstract** The past decade has seen the emergence of new diagnostics and drugs for tuberculosis, a disease that kills over 1.8 million people each year. However, these new tools are yet to reach scale, and access remains a major challenge for patients in low and middle income countries. Urgent action is needed if we are committed to ending the TB epidemic. This means raising the level of ambition, embracing innovation, increasing financial investments, addressing implementation gaps, and ensuring that new technologies reach those who need them to survive. Otherwise, the promise of innovative technologies will never be realized.

**MADHUKAR PAI*** **AND JENNIFER FURIN**

*For correspondence:
madhukar.pai@mcgill.ca

As the old saying goes, never bring a knife to a gunfight. And yet, the global tuberculosis (TB) community has been doing exactly that for decades – fighting a protracted battle with antiquated, inefficient tools (*Pai et al., 2016*), and losing millions of patients in the process.

The most commonly used TB test, sputum smear microscopy, dates back to 1882, the time Robert Koch first demonstrated the bacilli under a microscope. The Bacille Calmette–Guérin (BCG) vaccine, first used in humans in 1921, is still used globally, even though it has been largely ineffective in controlling the TB epidemic. On the treatment front, the TB community had to wait for nearly 50 years for new TB drugs to reach the market.

Lack of good tools has cost us dearly. Last year, the World Health Organization (WHO) declared that the TB epidemic was worse than previously thought, with an estimated 10.4 million new TB cases in 2015 (*World Health Organization, 2016a*). WHO also estimated that 1.8 million people died from TB in 2015, making TB a bigger killer than HIV and malaria combined.

Thanks to concerted efforts by donors, industry, product development partnerships, and other stakeholders, the past decade has seen the emergence of new diagnostics and drugs for TB. However, these new tools are yet to reach scale, and access remains a major challenge for patients in low and middle income countries (*Pai and Schito, 2015*; *Furin et al., 2016*).

## Access to new diagnostics

Several new TB diagnostics have emerged on the market and many have also been policy-endorsed by WHO and other agencies (*Pai and Schito, 2015*). These new diagnostics include Xpert® MTB/RIF (Cepheid Inc, Sunnyvale, California) (*World Health Organization, 2013a*), line probe assays (*World Health Organization, 2016b*), urine lipoarabinomannan antigen detection test (*World Health Organization, 2015a*), liquid cultures (*World Health Organization, 2007*), and interferon-gamma release assays (*World Health Organization, 2014a*).

Although it is widely acknowledged that rapid, accurate diagnosis is critical for timely initiation of anti-TB treatment, many people with TB (or symptoms of TB) struggle to access an adequate initial diagnosis. This is underscored by the fact that an estimated 41% of the 10.4 million new cases globally are either undiagnosed or not reported (*World Health Organization, 2016a*). These "missing" 4.3 million people with TB contribute to ongoing TB transmission, including the transmission of multidrug-resistant TB (MDR-TB).

Even if diagnosed with TB, access to universal drug-susceptibility testing is far from reality for

**Figure 1.** Cumulative number of GeneXpert instrument modules and Xpert MTB/RIF cartridges procured under concessional pricing since 2010. As of 31 December 2016, a total of 6,659 GeneXpert instruments (comprising 29,865 modules) and 23,140,350 Xpert MTB/RIF cartridges had been procured in the public sector in 130 of the 145 countries eligible for concessional pricing. Source: Cepheid & WHO (*World Health Organization, 2017a*).

most patients in high burden settings. Nearly 75% of the 480,000 cases of MDR-TB are either not detected or not reported (*World Health Organization, 2016a*). Even among previously treated patients at risk of drug-resistance and mortality, nearly 40% are not tested for drug resistance.

Cumulatively, since the launch of the Xpert MTB/RIF assay in 2010, over 6,500 GeneXpert machines and 23 million Xpert MTB/RIF cartridges had been procured in the public sector in 130 of the 145 countries eligible for concessional pricing, as of 31 December 2016 (*World Health Organization, 2017a*; *Figure 1*). While this trend is promising, the volume of Xpert MTB/RIF testing remains a small proportion of all TB tests conducted in high burden countries. It is estimated that over 77 million sputum smears are performed annually in 22 of the highest TB burden countries, including follow-up tests (*Kik et al., 2014*). With the notable exception of South Africa, which has rolled-out Xpert MTB/RIF on a national scale and accounts for almost 50% of global Xpert MTB/RIF volumes, other countries are still reliant on insensitive smears (*FIND et al., 2015*). This is reflected in the ratio of smear volumes to the number of Xpert MTB/RIF cartridges procured. In 2014, the ratio in South Africa was 1.6 – significantly lower

than most other high burden countries, where approximately 40–70 smears were performed for each Xpert MTB/RIF test (*Qin et al., 2015*). Although the ratio has become more favorable to Xpert MTB/RIF in 2015, smears still dominate the TB testing landscape in high burden countries (Danielle Cazabon, personal communication).

From a policy perspective, only eight countries have made Xpert MTB/RIF the initial diagnostic test for all people suspected of having TB, despite the high accuracy of the test and its ability to rapidly detect drug resistance (*Albert et al., 2016*). Even when the technology is available, access is limited to patients. A recent large, international study of Xpert MTB/RIF access and utilization in 18 countries found that the majority of sites had access to Xpert MTB/RIF, but only 4% of TB/HIV co-infected patients were actually tested using Xpert MTB/RIF (*Clouse et al., 2017*). Lack of diagnostics with high sensitivity, and inadequate access to drug-susceptibility testing forces patients to make multiple visits and spend a lot of money, and increases morbidity (*Chavan, 2017*).

Children are an especially vulnerable population when it comes to TB and MDR-TB. Although tools like Xpert have been endorsed for use in children (*World Health Organization, 2013a*), and Xpert has been shown to greatly improve the detection of MDR-TB in children (*Raizada et al., 2014*), access remains a concern in this subgroup.

Why are new TB diagnostics not reaching the patients who need it the most? As reviewed by Albert and colleagues (*Albert et al., 2016*), the roll-out of Xpert MTB/RIF has highlighted major implementation gaps that have constrained scale-up of this technology and limited its impact on patient outcomes. The roll-out has been hampered by high costs for cash-starved national TB control programs in high-burden countries. It is not just the cost of testing, but also the cost of treating MDR-TB, since Xpert MTB/RIF invariably identifies more patients with drug-resistance than conventional testing.

Another barrier for scale-up of Xpert MTB/RIF has been a lack of a complete diagnostic package for TB that includes comprehensive training, quality assurance, implementation plans, and maintenance support (*Albert et al., 2016*). Also, pragmatic trials have shown that the clinical impact of Xpert has been blunted by weak health systems, resulting in prolonged

time to diagnosis and treatment, compounded by empirical therapy by healthcare providers (*Theron et al., 2014*; *Durovni et al., 2014*; *Churchyard et al., 2015*).

In India, for example, an average TB patient is diagnosed after a delay of nearly two months and after seeing three providers (*Sreeramareddy et al., 2014*), and diagnosis is a major gap in the cascade of care in the public sector (*Subbaraman et al., 2016*) and the private sector (*Das et al., 2015*). In South Africa, even though Xpert MTB/RIF and line probe assays are freely available in centralized laboratories, studies show long delays between sample collection and initiation of TB treatment (*Hanrahan et al., 2012*; *Naidoo et al., 2014*), and missed opportunities to deploy tests that are available (*Chihota et al., 2015*).

In many countries the private sector plays a dominant role in care provision, yet this sector has been excluded from concessional Xpert MTB/RIF pricing, resulting in high costs for patients who pay an average price of nearly USD 70 for the test (*Puri et al., 2016*). In addition, quality of TB care, especially diagnosis, is a concern in the private sector (*Das et al., 2015*). Addressing this requires a comprehensive private sector engagement strategy that includes patient-centric quality care as a centerpiece (*Cazabon et al., 2017*).

In many countries, TB control receives limited domestic funding, and many low-income countries are heavily reliant on external donor funding (*Floyd et al., 2013*). India is a case in point. Despite having the largest number of TB patients in the world, the country's governmental expenditure on health is one of the lowest in the world at 1.4% of GDP (*The Lancet, 2017*). This under-investment is reflected in India's Revised National Tuberculosis Control Program (RNTCP), which has struggled to receive budgets necessary to tackle the world's biggest TB epidemic (*Pai et al., 2017*). This, in turn, has made India

# There is tension between wanting to "protect the new drugs" as opposed to protecting the lives of patients

rely heavily on old tools such as smear microscopy. Although over 600 GeneXpert systems are now available in the Indian public sector, and have been shown to dramatically increase the MDR-TB case detection rates (*Sachdeva et al., 2015*), the country has restricted the use of this technology primarily to individuals at risk of MDR-TB, persons with TB/HIV co-infection, and children. This might improve with India's new, ambitious plan to eliminate TB by 2025, although it is unclear whether India's TB control budget will match the ambition (*Pai et al., 2017*).

## Access to new drugs

Even if access to new diagnostics was to be dramatically improved, patients still need adequate access to drugs. The past five years have been a period of great potential with the introduction of two new therapeutic agents for the treatment of MDR-TB: delamanid (Otsuka Pharmaceutical Co, Tokyo, Japan) and bedaquiline (Janssen Therapeutics, NJ, USA) (*Gler et al., 2012*; *Diacon et al., 2014*; *Field et al., 2012*). Both drugs have been conditionally approved by stringent regulatory authorities and recommended by the WHO for the treatment of MDR-TB in situations in which there is resistance or intolerance to the other second-line agents or a high risk of treatment failure (*World Health Organization, 2013b*; *World Health Organization, 2014b*).

Although there was initial excitement about the availability of these new drugs, because they were approved using phase IIB data, their use was recommended in a cautious way (*World Health Organization, 2013b*; *World Health Organization, 2014b*, *World Health Organization, 2017b*). Since then, global introduction has not kept pace with the dire need for these drugs, especially for patients with drug-resistant disease who endure prolonged, toxic therapies with poor outcomes (*Lessem et al., 2015*). Complicating this have been the long delays seen in starting pivotal late-stage clinical trials for MDR-TB (*RESIST TB, 2017*).

A conservative estimate is that one-third of newly diagnosed patients with rifampicin-resistance will have an indication for a new drug during the treatment for MDR-TB, although some studies suggest that as many as two-thirds of patients would benefit from the new drugs (*Bonnet et al., 2016*). This means that there are between 200,000 and 400,000 persons in need of the novel medications each year. Even a more conservative estimate, based on a third of the

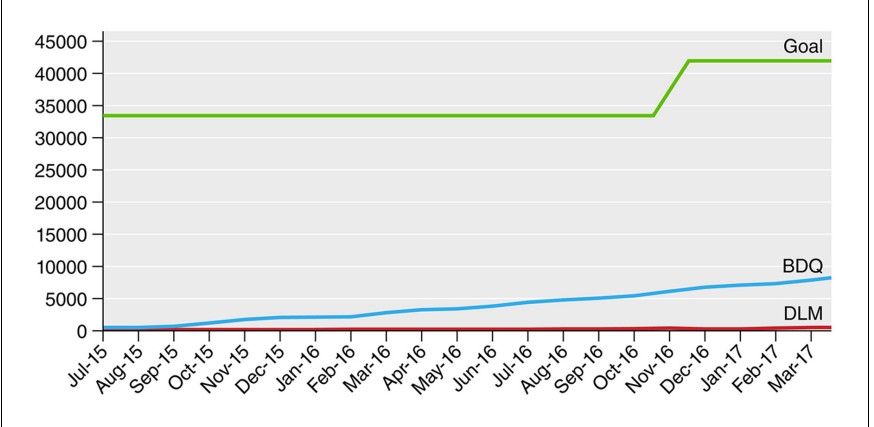

**Figure 2.** Progress in bedaquiline (BDQ) and delamanid (DLM) global uptake by month compared with estimated need (goal). Source: DR-TB STAT (*DR-TB STAT, 2017*).

total number of persons actually started on MDR-TB treatment each year, reveals a major access gap among the 41,667 patients who urgently need these medications each year.

Currently, just over 8,000 patients have ever received either bedaquiline or delamanid for the treatment of MDR-TB under program conditions (*DR-TB STAT, 2017*). Recent optimistic reports about new drug uptake released by the WHO – which state there are 70 countries using bedaquiline and 39 using delamanid – are somewhat misleading in that they count any country that has treated a single patient with a new drug, even under compassionate use conditions (*World Health Organization, 2016a*).

Program use data shows that as of March 1, 2017 there were 8,195 persons who have ever taken bedaquiline and 496 who have ever taken delamanid under program conditions (*DR-TB STAT, 2017*; *Figure 2*). Of note, 60% of global bedaquiline use is from South Africa (*DR-TB STAT, 2017*). So, most countries are either not using the new drugs or are focused only on small pilot projects. One example of this is in the Pan-American Health Organization (PAHO) region of the world, where it is estimated that there are more than 7,000 patients in need of either bedaquiline or delamanid, but fewer than 100 individuals who have received either of these medications.

There are several barriers for wider use of bedaquiline and delamanid (*Furin et al., 2016*). Some of these are related to costs and registration. Although there is a bedaquiline donation

program that allows Global Fund eligible countries to obtain bedaquiline free of charge, countries who are not eligible for this donation often pay a high price via a tiered pricing structure, with middle-income countries paying USD 3,000 for a six-month course of the drug, and high income countries paying as much as USD 26,000 for the six-month course – costs that are almost impossible to bear for poorly funded TB programs (*Gotham et al., 2017*). Some countries, however, have recognized that the cost of purchasing bedaquiline is lower than the cost to health systems of the ongoing spread of MDR-TB, as seen in the successful South African program. South Africa has to purchase its own bedaquiline yet it accounts for more than 60% of the global use (*Ndjeka, 2016*).

Delamanid is available for purchase at USD 1,700 for Global Fund eligible countries, but the costs in non-Global Fund eligible countries are excessive and may be as much as Euro 30,000 (approximately USD 32,000) in some settings (*Gotham et al., 2017*). While bedaquiline has been registered in a range of countries – including the high burden countries of South Africa and India – delamanid only has regulatory approval in four regions: European Union, Japan, South Korea, and Macau. None of these areas have a high burden of MDR-TB, and delamanid is still not registered in most of the countries where clinical trials were done.

Other important country-level barriers seem to be driving the slow uptake of new drugs on the part of national TB programs, regulatory agencies, and clinical providers. Some of these barriers are based in excessive concern about potential side effects, a concern that needs to be balanced with the high mortality rate in persons with poorly treated MDR-TB and the high rate of serious adverse events that occur using the existing medications (*Ahuja et al., 2012*). There is tension between wanting to "protect the new drugs" as opposed to protecting the lives of patients, with the drugs being restricted in order to purportedly preserve their efficacy. This was the situation in a recent court case where a young woman and her family sued the Indian government to provide her with access to bedaquiline – a case they won – but the government argued that doing so would risk the development of bedaquiline resistance (*Kirby, 2017*).

There has been almost no involvement of the private sector in the use of new drugs, even in private sector tertiary care facilities that have a proven track record in the successful

management of MDR-TB (*Udwadia et al., 2017*). There are also complicated pharmacovigilance requirements for new drugs (*World Health Organization, 2015b*) that have delayed their use in some settings (*Furin et al., 2016*). And most countries feel compelled to follow the interim WHO guidance to the letter on the use of these medications, and guidelines can lag behind the most relevant safety and efficacy data. Given the rapid accumulation of data on the safety and efficacy of these drugs, rigid adherence to interim policies can delay access to life-saving drugs for patients. It should be noted, however, that most countries are unable to implement new drugs even following these restrictive recommendations.

Children are especially vulnerable to TB and MDR-TB and they have been largely overlooked

## It is wonderful to see countries develop more ambitious plans for TB elimination, with India being a prominent recent example

when it comes to access to the new drugs, even though delamanid was recommended by the WHO for children as young as six years of age in 2016. Such exclusion is often a result of a desire to "protect" children from the unknown adverse effects of novel agents, a practice that is especially concerning given the well-known toxicity of the injectable agents (*Weld et al., 2017*). Not only is there a need to include children earlier in the clinical trials of novel agents (*Nachman et al., 2015*), but there is also a need to develop child-friendly formulations of the medications (*Furin et al., 2015*), something that is being pursued with both bedaquiline and delamanid. Other populations vulnerable to exclusion include pregnant women and persons with extrapulmonary TB. Thus, countries should make plans for off-label use of new drugs in these individuals, should the benefits outweigh the risks.

## Silver linings and potential solutions
Amidst the disappointingly slow pace of new tool uptake, there are some positive examples

and potential solutions. South Africa has been exceptional in scaling-up both Xpert MTB/RIF and bedaquiline, due in large part to a forward-thinking National TB Program and Department of Health, backed by an ambitious Health Minister, and supported by academic and non-governmental organization partners (*Ndjeka et al., 2015*). Other countries too have had success introducing bedaquiline, especially when they prioritized patient needs, were flexible with their innovations, and partnered with other supporting groups (*Guglielmetti et al., 2017*). Also, the early success of tools like Xpert MTB/RIF and bedaquiline have renewed industry engagement in TB R&D, pushed countries to develop systems for conducting field trials for policy changes, and revitalized front-line health workers and civil society (*Pai and Schito, 2015*; *Furin et al., 2017*). It is also wonderful to see countries develop more ambitious plans for TB elimination, with India being a prominent, recent example (*Central TB Division, 2017*).

Many of the barriers to optimal use of novel diagnostic and therapeutic strategies can be readily surmounted with bold ambitions that are followed up with concrete implementation. With regards to new diagnostics, we need a comprehensive approach to their implementation (*Albert et al., 2016*), including increased engagement of patients, civil society, and policy makers; demand creation for new tools among diverse stakeholders; broader health systems strengthening in preparation for new tools; increased advocacy for TB financing; and market-based approaches to address both demand and supply side barriers (*Engel et al., 2016*). These solutions are particularly relevant as WHO recently endorsed the use of Xpert MTB/RIF Ultra, a more sensitive assay than the current Xpert MTB/RIF cartridge (*World Health Organization, 2017c*), and point-of-care molecular diagnostics (e.g. GeneXpert Omni) are becoming a reality.

When it comes to the novel agents, while there are indeed limited data on their use in large populations, this should always be considered in the context of currently recommended treatment regimens, which are based on expert opinion and observational data, are associated with poor treatment outcomes, and are associated with an unacceptable safety profile that has been well documented. When viewed through this lens, the introduction of new therapeutic options can only be seen as a welcome event. The WHO must take this into account, and release unified MDR-TB guidelines that are

Pai and Furin. eLife 2017;6:e25956. DOI: 10.7554/eLife.25956

consistent in their use of recommendations based on observational and programmatic data. There has now been sufficient experience in the successful introduction of novel agents for MDR-TB, and the programs and providers responsible for this success should be leading the global roll-out of these drugs. Finally, urgent patient needs must be prioritized, with the organizations and agencies failing to address these needs being held accountable for delayed or denied access. After all, "protecting human rights, ethics and equity" is one of the four key principles of the WHO End TB Strategy (*World Health Organization, 2017d*).

## Conclusions

New diagnostic and treatment options have been sought for decades in the field of TB and MDR-TB, and new tools have been successfully introduced. But we have learnt the hard way that availability does not necessarily result in widespread access. Considerable effort is required to make sure new tools are made accessible to the persons who need them most. This has not happened as quickly as desired with tools such as Xpert MTB/RIF, bedaquiline, and delamanid, and the result is that tens of thousands of TB patients around the world suffer needlessly from late or misdiagnosis, ineffective treatment, or intolerable side effects. The mere fact that 1.8 million people die of a potentially curable infection each year is the surest indication that they are not getting the quality care they deserve.

Urgent action is needed if we are committed to ending TB in a mere 13 years. This means embracing innovation, increasing financial investments in TB control, addressing implementation gaps, and making sure that new technologies are available in the service of those who are trying to survive. More importantly, the global TB community needs to learn from the HIV/AIDS experience, and raise its level of ambition. This means a change in the mindset, to not settle for less.

**Madhukar Pai** is in the McGill Global Health Programs and McGill International Tuberculosis Centre, McGill University, Montreal, Canada, and the Manipal McGill Centre for Infectious Diseases, Manipal University, Manipal, India

http://orcid.org/0000-0003-3667-4536

**Jennifer Furin** is in the Department of Global Health and Social Medicine, Harvard Medical School, Boston, United States

*Author contributions:* MP, Involved in conceptualization of the article and contributed equally with JF to the writing, editing and revision of the article; JF, Involved in conceptualization of the article and contributed equally with MP to the writing, editing and revision of the article

*Competing interests:* MP: No industry or financial conflicts. MP is a consultant for the Bill & Melinda Gates Foundation (BMGF). BMGF had no involvement this manuscript. MP is an eLife Reviewing Editor. The other author declares that no competing interests exist.

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
