## [Decision Letter]

Thank you for submitting your article "Tuberculosis innovations mean little, if they cannot save lives" for consideration by eLife. Your article has been reviewed by 3 peer reviewers. The following individuals involved in review of your submission have agreed to reveal their identity: Jane Coyne (Reviewer #3).

I've copied the reviews below. In general, the reviewers were enthusiastic about the article, but they each have a number of points they would like you to address in a revised version of your article. Please try to address all these points in your revised version, or explain in the cover letter accompanying your resubmission why you did not feel it was essential to do so.

Reviewer #1:

1) Although there is nothing wrong in publishing this article as such, considering the authors' expertise, vast experience in the field of TB and other infectious diseases, they should also include few constructive suggestions on the above aspects than being generic.

2) Also, often time researchers ignore the necessity to concentrate more on the follow-up of TB cases started on treatment, and bacteriological confirmation at various time points to ensure the completion of treatment and disease free status. There is no diagnostic test available at present to differentiate dead and viable bacteria. Even the famous Bangladesh study relied only on vital staining techniques such as FDA/ EB based microscopy to prove that the MDR cases are not secreting non-viable bacilli. Considering this need, the authors must urge the scientific community to concentrate on this aspect as well.

3) People generally tend to ignore paediatric and extra-pulmonary TB cases due to diagnostic constraints. Although XPert does wonders in the early detection of such cases, access becomes difficult and also specimen collection, which requires additional provision including hospitalization, negates the optimal utility of XPert. With any new technique when the sensitivity gets enhanced, the danger of quality being compromised lingers on. To overcome this, the necessity to establish quality assurance in public and private care should be included.

4) Likewise, the easy access and availability of drugs in suitable formulations for paediatric TB cases should be advocated. This applies more so for newer drugs.

5) The upcoming improved version of Xpert such as XPert Ultra, Omni and other methods such as TB LAMP, True NAT etc may also be mentioned to show that some hope is on the horizon.

6) Finally TB control should be inclusive and we are all aware that involving private care remains mostly on paper now for various reasons. Until the respective governments give their commitment by way of legislation or decree, we will be rendering mostly lip service and any amount of writing may not solve the problem. This may be advocated in the UN Assembly as and when they hold a special session on TB Control. Can this be considered by the authors and also other points submitted.

Reviewer #2:

1) The authors omit to mention that Bdq and Dlm are not fully approved by any SRA but got either accelerated or conditional approval by FDA or EMA after their phase IIb. This justifies the pharmacovigilance requirements and partly explains the reluctance of some countries to use these drugs.

2) The paper fails to recognize and describe the major obstacles that lie at country level (regulatory authorities, NTPs, or reluctance from clinicians): India which is eligible to GF is limiting access to Bdq to 6 centers of excellence for a country of 1.3 billion inhabitants; Peru, despite registration and free access to Bdq has only enrolled 71 patients, Philippines 51, etc.

3) Presenting the cost as a major obstacle to access is somehow contradicted (or at least tempered) by the facts: there is very little uptake of Bdq by most GF eligible countries for which Bdq is free (numerous examples: Peru, Kyrgyzstan, Bangladesh, India, etc.) when South Africa (which pays Bdq at 900 USD per treatment because of internal procurement mechanism) is far ahead with over 4,000 treatments.

4) There is no doubt that the WHO recommendations on new drugs should be revised, however most countries are far from even reaching the current WHO recommended indications. One could quite easily make estimates showing that following these recommendations "to the letter", as the authors say, would rather result in a much greater uptake than what is actually observed.

5) Minor comments

-77 million sputum smears is a bit misleading as it includes the follow-up smears

- Presenting the costs of the new drugs in high income countries is not relevant to the scope of the paper

- I would suggest referencing the WHO guidelines for Xpert, LPA, LAM, etc.

- Bonnet et al. found that 2/3 of their cohort (mainly from CIS countries) could benefit from the new drugs

- Presenting the WHO recommended basic aDSM as a complex pharmacovigilance slowing down the uptake seems quite exaggerated.

*Reviewer #3*:

1) Figure 1 - I think it would be stronger if you added graphically to it the estimated number of smear tests - to show the GeneX growth in the context of smear usage

2) Diagnostics section - when you mention private sector I think you are wasting an opportunity - the issue is not just with concessionary pricing it's that there is no effective oversight and no political leadership to assure quality of diagnosis and care in the private sector.

3) Paragraph three - I would calculate the estimated number of patients that need access to these drugs (480K *1/3 = 160K annually) to emphasize how far off the mark we are with 7000 ttt's total. This would strengthen the deficit

---

## [Author Response]

Reviewer 1

1) Although there is nothing wrong in publishing this article as such, considering the authors' expertise, vast experience in the field of TB and other infectious diseases, they should also include few constructive suggestions on the above aspects than being generic.

Thank you. We agree and have provide several constructive suggestions towards the end of the revised version. Please see the revised section on S*ilver linings and potential solutions*.

2) Also, often time researchers ignore the necessity to concentrate more on the follow-up of TB cases started on treatment, and bacteriological confirmation at various time points to ensure the completion of treatment and disease free status. There is no diagnostic test available at present to differentiate dead and viable bacteria. Even the famous Bangladesh study relied only on vital staining techniques such as FDA/ EB based microscopy to prove that the MDR cases are not secreting non-viable bacilli. Considering this need, the authors must urge the scientific community to concentrate on this aspect as well.

Our article was about low access to TB innovations that already exist. We agree that a test for cure is necessary, but we would like to avoid listing all the unmet needs in TB, since that is not the goal of our paper.

3) People generally tend to ignore paediatric and extra-pulmonary TB cases due to diagnostic constraints. Although XPert does wonders in the early detection of such cases, access becomes difficult and also specimen collection, which requires additional provision including hospitalization, negates the optimal utility of XPert. With any new technique when the sensitivity gets enhanced, the danger of quality being compromised lingers on. To overcome this, the necessity to establish quality assurance in public and private care should be included.

We have added the following text to address this:

‘Children are an especially vulnerable population when it comes to TB and MDR-TB and although tools like Xpert have been endorsed for use in children (WHO policy), and Xpert has been shown to greatly improve the detection of MDR-TB in children (Raizada et al), access remains a concern in this subgroup.’

4) Likewise, the easy access and availability of drugs in suitable formulations for paediatric TB cases should be advocated. This applies more so for newer drugs.

Children have been virtually overlooked when it comes to access to the new drugs, even though delamanid was recommended by the World Health Organization for children as young as six years of age in 2016 (World Health Organization, 2016). Such exclusion is often a perverse result of a desire to “protect” children from the unknown adverse events of novel agents, a practice that is especially concerning given the well-known toxicity of the injectable agents (Weld, E., et al., 2017). Not only is there a need to include children earlier in the clinical trials of novel agents (Nachman et al., 2015) but also to develop child-friendly formulations of the medications (Furin et al., 2015), something that is being pursued with both bedaquiline and delamanid. Other populations vulnerable to exclusion include pregnant women and persons with extrapulmonary TB, and countries should make plans for off-label use of new drugs in these individuals should the benefits outweigh the risks.

5) The upcoming improved version of Xpert such as XPert Ultra, Omni and other methods such as TB LAMP, True NAT etc may also be mentioned to show that some hope is on the horizon.

We have added a note about the recent WHO endorsement of Xpert Ultra and other newer tools.

6) Finally TB control should be inclusive and we are all aware that involving private care remains mostly on paper now for various reasons. Until the respective governments give their commitment by way of legislation or decree, we will be rendering mostly lip service and any amount of writing may not solve the problem. This may be advocated in the UN Assembly as and when they hold a special session on TB Control. Can this be considered by the authors and also other points submitted.

We have added the following text to the article:

‘There has been almost no involvement of the private sector in the use of new drugs, even in private sector facilities that have a proven track record in the successful management of MDR-TB (Udwadia et al., 2016).’

Reviewer 2

1) The authors omit to mention that Bdq and Dlm are not fully approved by any SRA but got either accelerated or conditional approval by FDA or EMA after their phase IIb. This justifies the pharmacovigilance requirements and partly explains the reluctance of some countries to use these drugs.

We added the term “conditionally approved” to the manuscript.

2) The paper fails to recognize and describe the major obstacles that lie at country level (regulatory authorities, NTPs, or reluctance from clinicians): India which is eligible to GF is limiting access to Bdq to 6 centers of excellence for a country of 1.3 billion inhabitants; Peru, despite registration and free access to Bdq has only enrolled 71 patients, Philippines 51, etc.

Most of the barriers mentioned here are discussed in the text. However, in order to emphasize their importance, we have added the following sentence:

Other important country-level barriers seem to be driving the slow uptake of new drugs on the part of regulatory agencies, NTPs, and clinical providers.

3) Presenting the cost as a major obstacle to access is somehow contradicted (or at least tempered) by the facts: there is very little uptake of Bdq by most GF eligible countries for which Bdq is free (numerous examples: Peru, Kyrgyzstan, Bangladesh, India, etc.) when South Africa (which pays Bdq at 900 USD per treatment because of internal procurement mechanism) is far ahead with over 4,000 treatments.

We thank the reviewer for pointing this out. We have added the following sentence to the paragraph to show that cost is not the only barrier to bedaquiline use.

Some countries, however, have recognized that the cost of purchasing bedaquiline is lower than the health systems costs of ongoing spread of MDR-TB, as seen in the successful South African program. South Africa has to purchase its own bedaquiline yet it accounts for more than 60% of the global use (Ndjeka et al., 2016)

4) There is no doubt that the WHO recommendations on new drugs should be revised, however most countries are far from even reaching the current WHO recommended indications. One could quite easily make estimates showing that following these recommendations "to the letter", as the authors say, would rather result in a much greater uptake than what is actually observed.

We have added the following statement in to address the reviewer’s concerns:

‘It should be noted, however, that most countries are unable to implement new drugs even following these restrictive recommendations.’

5) Minor comments

-77 million sputum smears is a bit misleading as it includes the follow-up smears

We have added a note that this includes follow-up smears.

- Presenting the costs of the new drugs in high income countries is not relevant to the scope of the paper

We disagree with this, as the high costs have greatly limited access to the new drugs in wealthier countries, and they are reserved only for the sickest patients. Pricing is an issue and we feel it is important to mention the range of prices.

- I would suggest referencing the WHO guidelines for Xpert, LPA, LAM, etc.

We have added these references

- Bonnet et al. found that 2/3 of their cohort (mainly from CIS countries) could benefit from the new drugs

We have added the following text to the paper to address the reviewer’s concerns: although some studies suggest that as many as two-thirds of patients would benefit from the new drugs (Bonnet, Bastard et al. 2016)

- Presenting the WHO recommended basic aDSM as a complex pharmacovigilance slowing down the uptake seems quite exaggerated.

While we appreciate the reviewer’s opinion on this topic, data on early barriers to using new drugs was collected as part of the work reported in the Furin et al., 2016 article. A majority of the countries named the need for active PV as a significant barrier to introduction. Most TB programs use passive reporting systems, and in some countries, no PV system existed at all. Because this was reported by countries as a major reason for not introducing the new drugs, we prefer to keep it in the paper, even if the reviewer feels it is an exaggeration.

Reviewer 3

*1) Figure 1 - I think it would be stronger if you added graphically to it the estimated number of smear tests - to show the GeneX growth in the context of smear usage*

We have a separate paper in preparation, on the Smear to Xpert cartridge ratio across 22 HBCs. Since that work is not published, we are not able to add the estimated smear numbers to the existing Fig 1.

2) Diagnostics section - when you mention private sector I think you are wasting an opportunity - the issue is not just with concessionary pricing it's that there is no effective oversight and no political leadership to assure quality of diagnosis and care in the private sector.

We agree and have noted this in the revised version:

In addition, quality of TB care, especially diagnosis, is a concern in the private sector (Das, Kwan et al. 2015). This requires a comprehensive private sector engagement that includes quality improvement strategies (Cazabon, Alsdurf et al. 2017).

*3) Paragraph three - I would calculate the estimated number of patients that need access to these drugs (480K *1/3 = 160K annually) to emphasize how far off the mark we are with 7000 ttt's total. This would strengthen the deficit*

DR-TB STAT, the organization whose work is cited here, takes a more conservative approach to estimating new drug needs, and they base it not on the estimated 580,000 persons with RR-TB each year, but rather on the numbers of persons actually started on treatment. In order to address this concern, we have added the following statement:

although some studies suggest that as many as two-thirds of patients would benefit from the new drugs (Bonnet, Bastard et al. 2016)—meaning that there are between than 200,000 and 400,00 persons in need of the novel medications. Even a more conservative estimate based on the number of persons actually started on MDR-TB treatment each year reveals a major access gap among the 41,667 patients who urgent need these medications.

Overall in this section I think you are missing a major part of the narrative. In my opinion, the TB community has collectively failed to get the Phase 3 trials through in a timely manner. It's a little tough to say that the guidelines haven't changed when the pool of data (except observational cohort data) hasn't really moved in 4 years. There are many contributory factors but this is super slow - money, trials sites, regimen design, data access issues yes - but this is another strong signal of the lack of ambition.

We have added the following sentence to the paper to address the reviewer’s concern: Complicating this have been the long delays seen in starting pivotal later stage clinical trials for MDR-TB (RESIST TB, 2017).